# Mitochondrial Protease Oct1p Regulates Mitochondrial Homeostasis and Influences Pathogenicity through Affecting Hyphal Growth and Biofilm Formation Activities in *Candida albicans*

**DOI:** 10.3390/jof10060391

**Published:** 2024-05-30

**Authors:** Xiaoxiao Zhu, Feng Jin, Guangyuan Yang, Tian Zhuang, Cangcang Zhang, Hanjing Zhou, Xiaojia Niu, Hongchen Wang, Daqiang Wu

**Affiliations:** 1Department of Pathogenic Biology and Immunology, College of Integrated Chinese and Western Medicine, Anhui University of Chinese Medicine, 350 Longzihu Road, Hefei 230012, China; 2Key Laboratory of Xin’an Medicine, Ministry of Education, College of Nursing, Anhui University of Chinese Medicine, 350 Longzihu Road, Hefei 230038, China

**Keywords:** *Candida albicans*, Oct1p, mitochondria, hypha formation, virulence

## Abstract

Mitochondria, as the core metabolic organelles, play a crucial role in aerobic respiration/biosynthesis in fungi. Numerous studies have demonstrated a close relationship between mitochondria and *Candida albicans* virulence and drug resistance. Here, we report an octapeptide-aminopeptidase located in the mitochondrial matrix named Oct1p. Its homolog in the model fungus *Saccharomyces cerevisiae* is one of the key proteins in maintaining mitochondrial respiration and protein stability. In this study, we utilized evolutionary tree analysis, gene knockout experiments, mitochondrial function detection, and other methods to demonstrate the impact of Oct1p on the mitochondrial function of *C. albicans*. Furthermore, through transcriptome analysis, real-time quantitative PCR, and morphological observation, we discovered that the absence of Oct1p results in functional abnormalities in *C. albicans*, affecting hyphal growth, cell adhesion, and biofilm formation. Finally, the in vivo results of the infection of *Galleria mellonella* larvae and vulvovaginal candidiasis in mice indicate that the loss of Oct1p led to the decreased virulence of *C. albicans*. In conclusion, this study provides a solid theoretical foundation for treating Candida diseases, developing new targeted drugs, and serves as a valuable reference for investigating the connection between mitochondria and virulence in other pathogenic fungi.

## 1. Introduction

*Candida albicans* is a fungus with conditional pathogenicity that can be found inhabiting various regions of the human body, such as the skin, mucosa, digestive tract, and other organs [1]. Infections caused by *C. albicans* range from superficial conditions to systemic diseases [2]. In individuals with a functioning immune system, *C. albicans* is able to coexist harmoniously with the body’s microflora without posing a threat to the host’s health. Conversely, in cases where the host’s immune function is compromised and the flora is imbalanced, *C. albicans* has the potential to transition to a pathogenic state and induce infection in the host [3]. The comparison of gene function between *C. albicans* and *Saccharomyces cerevisiae* reveals notable differences. Investigating the gene function of *C. albicans* is crucial for comprehending its biological traits, as well as its pathogenicity, resistance mechanisms, and molecular interactions with host cells. These studies offer valuable insights for the development of treatments for invasive fungal infections.

Mitochondria are distinct organelles in eukaryotic cells that primarily serve to supply energy to the cell through the process of respiration [4,5]. At the same time, mitochondria serve as central hubs in fungal cells to coordinate a variety of cell biological functions, such as metabolite production, biofilm formation, drug resistance, and virulence [6]. The function of mitochondria plays a crucial role in fungi’s (Aspergillus fumigatus, Cryptococcus neoformans) ability to cause disease, particularly when they adapt to challenges in the mammalian environment (e.g., nutrient limitations, stress responses, and immune responses) [7,8,9]. Numerous studies have demonstrated the significance of mitochondria as crucial organelles in *C. albicans* pathogenesis and their close association with the drug resistance mechanisms of this fungal pathogen [10]. Mitochondria are essential for the transition of *C. albicans* from the yeast form to the hyphal form, as well as for cell wall biosynthesis, drug resistance, and virulence [11]. Mitochondrial proteases refer to proteases and peptidases that are expressed in mitochondria or transported to mitochondria post-translation. They play a crucial role in various aspects of mitochondrial function [12]. Additionally, these proteases are involved in the activation and deactivation of proteins essential for the core mitochondrial functions in a highly specific and regulated manner, exerting significant influence [13]. Simultaneously, mitochondrial proteases are intricately associated with cellular senescence, and their impairment may play a role in the pathogenesis of diverse pathological states, including cancer, metabolic syndrome, and neurodegenerative disorders [14].

In *S. cerevisiae*, Oct1p functions as an octapeptide-aminopeptidase located within the mitochondrial matrix. Following cleavage of the mitochondrial targeting sequence (MTS) by mitochondrial processing protease (MPP), Oct1p catalyzes the removal of eight amino acids from the N-terminal region of the specific protein, thereby facilitating the conversion of unstable precursor intermediates produced by MPP into stable mature proteins [15]. Additionally, the deletion of the *OCT1* gene is expected to cause mitochondrial DNA loss, resulting in respiratory impairment [16]. This study involved the knockout of the *OCT1* gene in *C. albicans*, revealing a notable influence on both biofilm formation and hyphal growth. Despite the significance of *C. albicans* as a pathogen, the role of Oct1p in its virulence remains uncertain.

The objective of this study was to examine the impact of Oct1p on the virulence of *C. albicans*. We demonstrated that Oct1p not only affects the carbon utilization capacity and temperature sensitivity of *C. albicans* but also reduces the virulence of *C. albicans*. The *oct1*∆/∆ mutant displayed notable mitochondrial dysfunction, hindered biofilm formation, and impaired hyphal development. Furthermore, transcriptome sequencing analysis revealed a number of down-regulated genes associated with hyphal formation, cell aggregation, and adhesion. This research also investigated the transcriptional expression levels of hyphal growth and adhesion-related genes (*ECE1*, *FTR1*, *ALS1*, *ALS3*) in relation to *OCT1* deletion. Additionally, it evaluated the infectivity of *G. mellonella* and the localized infectivity in mice. The findings demonstrate that Oct1p plays a role in regulating hyphal growth, pathogenicity, and interaction with the host.

## 2. Materials and Methods

### 2.1. Strain and Growth Conditions

The wild-type strain SN152 and plasmids pSN40 and pSN52 used in this study were provided by Prof. Yuanying Jiang from the Center for New Drug Research at the School of Pharmacy, Second Military Medical University. Additionally, plasmid CIp30 was provided by Prof. Alistair J.P. Brown from the University of Exeter.

The strains utilized in this investigation are outlined in Appendix A [17]. *C. albicans* strains were cultivated at 30 °C in a yeast extract peptone dextrose medium (1% yeast extract, 2% peptone, 2% dextrose, YPD liquid medium) and yeast extract peptone dextrose agar medium (1% yeast extract, 2% peptone, 2% dextrose, and 2% agar, YPD solid medium) for general growth and propagation. The strains were preserved in a solution containing 50% glycerol at a temperature of −80 °C and were acquired through incubation at a temperature of 30 °C on YPD solid plates.

### 2.2. Strains’ Construction

The *C. albicans* strain SN152 is characterized by mutations in the leucine (*LEU*), histidine (*HIS*), and arginine (*ARG*) genes [17]. This study employed the *HIS1-LEU2-ARG4* nutrient-deficiency method and homologous recombination principles to generate the *oct1*∆/∆ mutant and the *OCT1* gene-reconstituted strain *oct1*∆/∆:*OCT1*. Appendix A provides a comprehensive listing of all primers employed in the strain construction process, while Appendix A [17,18] offers a summary of all plasmids utilized. The *HIS1-LEU2-ARG4* nutrient-deficient assay is commonly employed for the sequential knockout of target genes, with the flexibility to select two screening markers for this purpose. In this study, the *HIS1* cassette from plasmid pSN52 was amplified, and the strain SN152 was transformed with the fusion PCR products of *HIS1* flanked by *OCT1* 5′ and 3′ fragments. Plasmid pSN40 was used as a PCR template to amplify the *LEU2* cassette, and the second *OCT1* allele was deleted using fusion PCR products of the *LEU2* marker flanked by *OCT1* 5′ and 3′ fragments. In this research, the open reading frame (ORF) of the Oct1p protein was integrated into the RPS1 site of the CIP30-Num11 promoter recombinant plasmid, which also contained the heterologous screening marker *C. dubliniensis ARG4*. This was achieved through homologous recombination using the ectopic backfill method. Subsequently, the constructed plasmid was transferred into the *oct1*∆/∆ mutant via *Stu I* digestion, resulting in the *OCT1* gene-reconstituted strain *oct1*∆/∆:*OCT1*. The construction process was successfully concluded, with all strains generated in this research project being identified through sleeve polymerase chain reaction (PCR) analysis, as depicted in Appendix A.

### 2.3. Phenotyping of Strains

In this study, the *oct1*Δ::*HIS1*, *oct1*Δ/Δ, *oct1*∆/∆:*OCT1* strains and WT were cultured in various nutrient-deficient media (SC-His, SC-Leu-His, and SC-Leu-His-Arg) alongside a control group cultured in YPD. The growth of the strains was monitored after 48 h of incubation at 30 °C.

### 2.4. Mitochondrial Morphology

In this study, the mitochondrial morphology was examined using Mito-Tracker Red fluorescent dye (Beyotime, Shanghai, China) [19,20]. Initially, strains were cultured overnight in YPD, and then the concentration of *C. albicans* suspension was adjusted to 1.0 × 10^6^ cells/mL by phosphate buffered saline (PBS). Subsequently, 200 μM of Mito-Tracker Red dye solution was introduced into the *C. albicans* suspension at a ratio of 1:1000, and the mixture was incubated at 30 °C for 30 min in the dark. The suspension was then washed twice with PBS in order to observe the mitochondrial morphology under excitation at a wavelength of 579 nm (Leica DMi8, Hesse, Germany).

### 2.5. Measurement of Intracellular ATP Contents

To investigate the impact of Oct1p protein depletion on the intracellular ATP levels of the strains, the BacTiter-Glo^TM^ Microbial Cell Activity Assay Kit (Promega, Madison, WI, USA) was employed for quantifying the intracellular ATP content [21]. Strains were cultured overnight in YPD. Subsequently, the concentration of the *C. albicans* suspension was adjusted to 1.0 × 10^7^ cells/mL in PBS. An equal volume of the *C. albicans* suspension was then incubated with BacTiter-Glo reaction solution for 5 min. The resulting RLU fluorescence signals were detected using a full-wavelength multifunctional enzyme labeling instrument (Thermo Fisher Scientific, Shanghai, China).

### 2.6. Measurement of Mitochondrial Membrane Potential

To assess the mitochondrial membrane potential of each strain, we utilized the JC-1 Mitochondrial Membrane Potential Assay Kit (Beyotime, Shanghai, China) to determine the mitochondrial membrane potential level [22]. Strains were cultured overnight in YPD. Subsequently, the concentration of the *C. albicans* suspension was adjusted to 1.0 × 10^7^ cells/mL in PBS. Subsequently, an equal volume of *C. albicans* suspension was combined with JC-1 staining solution and incubated at 37 °C for 20 min in the dark. The mixture was then washed twice with PBS. The fluorescence intensity at emission wavelengths of 590 nm and 530 nm was concurrently measured using a flow cytometer (BD, Franklin Lakes, NJ, USA), and the red/green fluorescence intensity ratio was documented for each sample.

### 2.7. Measurement of Intracellular Reactive Oxygen Species Levels

To examine the impact of *OCT1* deletion on the intracellular reactive oxygen species (ROS) levels of *C. albicans*, the Reactive Oxygen Assay Kit (Nanjing Jiancheng, Nanjing, China) was utilized to measure ROS content within each strain. Cultures of strains were grown in YPD liquid medium overnight at 30 °C with shaking at 200 rpm. Cells were adjusted to 1.0 × 10^6^ cells/mL with PBS. DCFH-DA was added to each solution at 10 μM and incubated at 30 °C for 1 h. After rinsing with PBS, visualization was performed using a fluorescence microscope (Olympus, Tokyo, Japan) with consistent exposure settings at an excitation wavelength of 488 nm and an emission wavelength of 525 nm. The fluorescence intensity was quantified utilizing Image J 1.8.0 software.

### 2.8. Susceptibility Assays

To assess strain growth under various stresses, strains were first cultured in YPD overnight. The cells were then adjusted to 1.0 × 10^6^ cells/mL in PBS and further diluted to 1.0 × 10^2^ cells/mL at a 10-fold ratio. *C. albicans* samples were spotted with 5 μL on plates containing different carbon sources and stress reagents, and then incubated at various temperatures for 48 h to observe growth. In addition, we diluted cells to 300 cells/mL in PBS, applied 100 μL to YPD and YPEG solid medium, and incubated at 30 °C, 35 °C, and 37 °C for 48 h [23]. Colony size was measured with vernier calipers at the end of the incubation.

The fermentation carbon sources included 2% glucose (MACKLIN, Shanghai, China), 2% maltose (MACKLIN, Shanghai, China), and the non-fermentation carbon sources included 2% ethanol (Jiangsu Qiangsheng Functional Chemistry Co., Ltd., Suzhou, China), sodium citrate (Sinopharm Pharmaceutical Group Pharmaceutical Co., Ltd., Beijing, China), and 2% glycerol. The carbon source and temperature synergies included a fermentation carbon source (YPD): 2% glucose, and a respiration carbon source (YPEG): 2% glycerol and anhydrous ethanol mixture. Temperatures included 30 °C, 35 °C, and 37 °C. The cell membrane stress reagents included 2 μg/mL fluconazole (FLU) (Shanghai yuanye Biotechnology Co., Ltd., Shanghai, China), 0.05 μg/mL Amphotericin B (AmB) (Shanghai yuanye Biotechnology Co., Ltd., Shanghai, China), and 0.04% sodium dodecyl sulfate (SDS) (Coolaber, Beijing, China) [24]. The cell wall stress components included 50 μg/mL Congo Red (CR) (Solarbio, Shanghai, China) and 200 μg/mL Calcium fluorescent white (CFW) (Sigma, Saint Louis, MO, USA) [25].

### 2.9. Growth Curve Measurement

To assess strain growth, we incubated them in YPD liquid medium overnight until they reached a stationary phase. Then, we adjusted the optical density (OD) to 0.04 at 600 nm with sabouraud dextrose broth (SDB) liquid medium after washing with PBS. *C. albicans* suspensions were then incubated at 30 °C with shaking at 200 rpm for 24 h. The OD_600_ values of *C. albicans* suspensions were measured every 2 h, and the growth curve of *C. albicans* suspensions was plotted at different time points. The doubling time of *C. albicans* cells during the logarithmic growth period was calculated according to the concentration of *C. albicans* suspensions, and the formula was as follows: doubling time (DT) = t × lg2 ÷ lg[N_t_ ÷ N_0_], where N_t_ is the number of bacterial cells at t time, and N_0_ is the initial number of *C. albicans* cells [26].

### 2.10. RNA-Seq Analysis

To prepare samples, all strains were incubated in YPD liquid medium at 30 °C overnight. Subsequently, the overnight culture was inoculated into fresh YPD for further cultivation, and the cells were harvested at the log phase (OD_600_ = 0.5–0.8) for cryopreservation in liquid nitrogen (3 samples were repeated for each group). The work of transcriptome sequencing was performed by Tsingkebio Company, Beijing, China.

### 2.11. Quantitative Real-Time PCR

All the strains were cultured overnight in YPD at 30 °C and subcultured in fresh YPD (buffered) at 30 °C until the log phase (OD_600_ = 0.5–0.8). Subsequently, we used a SPARKeasy Yeast RNA Extraction Kit (SPARKeasy, Jinan, China) to extract RNA (3 samples were taken from each group). In order to assess the transcript levels of vaginal inflammatory factors in the various groups of mice, RNA was extracted from mouse vaginal tissues utilizing the Trizol method (3 samples were taken from each group). In summary, mouse vaginal tissue was isolated and homogenized in a precooled mortar with liquid nitrogen. The resulting sample was then mixed with 1 mL Trizol and incubated for 10 min. Subsequently, chloroform (200 µL) was added to facilitate the separation of the aqueous RNA-containing solution from the organic layers, followed by RNA precipitation using isopropyl alcohol (equal volume) and the addition of 1 mL ethanol (75%) to each RNA pellet. The samples were then centrifuged (12,000 rpm/5 min/4 °C), and the RNA was resuspended in 20 mL nuclease-free water for storage [2]. The ToloScript ALL-in-one RT EasyMix for qPCR kit (TolobioL, Shanghai, China) was utilized for reverse transcription of the proposed RNA into cDNA. Then, qRT-PCR was performed using the 2 × Q3 SYBR qPCR Master Mix kit (Tolobio, Shanghai, China) using cDNA as the template and β-actin as the housekeeping gene used as the control. Gene amplification was determined using the 2^−ΔΔCT^ method. The primers for the genes used in the qRT-PCR are shown in Appendix A.

### 2.12. Filamentous Growth Assays

All strains were cultured overnight in YPD, followed by washing and dilution with PBS to achieve a suspension with 1.0 × 10^5^ cells/mL. The adjusted *C. albicans* suspensions were spotted to YPD, YPD + 10% FBS, Lee’s, Spider, and SLAD plates using 5 μL of each solution, followed by incubation at 37 °C for 7 days. Subsequently, the development of hyphae was monitored. Furthermore, the cells were subsequently standardized to a concentration of 1.0 × 10^6^ cells/mL using six different liquid media known to induce hyphal growth: RPMI-1640, YPD, YPD + 10% FBS, Lee’s, Spider, and SLAD [27,28,29]. The standardized *C. albicans* suspensions were then incubated at 37 °C for 4 h, after which hyphal growth was examined under a microscope.

### 2.13. Biofilm Formation Assay

To test biofilm formation, we incubated strains in YPD medium overnight, then adjusted *C. albicans* concentration to 1.0 × 10^6^ cells/mL in RPMI-1640. An amount of 100 μL of each solution was added to a 96-well plate, incubated at 37 °C for 90 min, and washed with PBS. The supernatant was removed and subjected to two washes with PBS before being supplemented with 100 μL of fresh RPMI-1640 liquid medium in each well. The mixture was gently agitated and subsequently incubated at 37 °C for 24 h. Following this incubation period, the biofilm was washed twice with PBS. Subsequently, the supernatant was aspirated, and 150 μL of XTT working solution was added before measuring the absorbance value of each well at 490 nm following a 3 h incubation at 37 °C [30].

We also used the CFW staining method to study biofilm formation. Strains were incubated in YPD liquid medium overnight, then adjusted to 1.0 × 10^6^ cells/mL in RPMI-1640 and incubated for 24 h at 37 °C [31]. The cells were washed with PBS and stained with 50 μg/mL CFW before observing biofilm formation under a microscope.

### 2.14. In Vitro Adhesion Assay

To assess alterations in the in vitro adhesion capacity of *C. albicans* following the deletion of the *OCT1* gene, fetal bovine serum (FBS) (Kangyuan Biotechnology Co., Ltd., Tianjin, China) was introduced into 12-well plates and incubated at 37 °C with agitation at 75 rpm overnight. Subsequently, each strain was cultured in YPD liquid medium at 30 °C overnight, followed by two washes with PBS and adjustment of the *C. albicans* concentration to OD_600_ = 0.5 using Spider liquid medium. Fetal bovine serum was removed from the 12-well plate, washed with PBS, and then *C. albicans* suspensions were added and incubated at 37 °C, 150 rpm for 48 h. The supernatant was then removed, unadhered cells were washed with PBS, and the results were observed.

### 2.15. Galleria Mellonella Infection Experiment

To examine the impact of *OCT1* deletion on infection in vivo, a total of 72 sensitive larvae measuring 2.0–2.5 cm in body length were chosen for assessing the virulence of each strain. All strains were cultured overnight in YPD at 30 °C and adjusted to 8.0 × 10^6^ cells/mL in PBS. The *G. mellonella* were subsequently divided into four groups: Blank group, WT group, *oct1*Δ/Δ group, and *oct1*∆/∆:*OCT1* group. Each group received a 10 µL injection of the corresponding fungal solution into the left or right hypophysis using a pointed microinjector needle. The *G. mellonella* were then incubated at 37 °C for 48 h. After incubation, three larvae from each group were randomly selected, ground, and plated on YPD solid plates. The plates were then inverted and incubated at 30 °C for an additional 48 h. The fungal load results were analyzed for each group. Furthermore, the mortality rates of *G. mellonella* were monitored at 24 h intervals over a period of 14 days, and survival curves were subsequently generated [32].

### 2.16. Experimental Vulvovaginal Candidiasis (VVC) Infection in Mice

In order to examine the impact of *OCT1* deletion on localized infection in vivo, a cohort of 48 SPF-grade female Kunming mice (weighing 20 ± 2 g) were chosen and acclimated for 7 days with a standard pellet diet and tap water under conditions of 12 h of light and 12 h of darkness at a temperature of 23 ± 2 °C and a relative humidity of 40–70%. The maintenance and treatment of all animals complied with the regulations of the Animal Ethics Committee of the Chinese Center for Disease Control and Prevention and the “Guidelines for the Care and Use of Laboratory Animals” in China.

During the second week of the experiment, the mice were allocated into four groups—Control, WT, *oct1*Δ/Δ, and *oct1*∆/∆:*OCT1*—using a random assignment method. Subsequently, all mice received subcutaneous injections in the neck on days 1, 3, and 5 with 0.1 mL of estradiol benzoate at a concentration of 0.1 mg/mL dissolved in sterile edible oil (dibai Biotechnology Co., Ltd., Shanghai, China). Three injections were administered to induce pseudo-estrus in mice, followed by inoculation with 20 µL of *C. albicans* suspension (2 × 10^7^ cells/mL) in the vagina for vaginal infection on the day after the final injection [33]. The control group received 20 µL of PBS for vaginal stimulation. The mice were monitored daily over a period of seven consecutive days to assess any morphological changes in the vaginal orifices and to evaluate their overall health status. On the seventh day post-infection, a vaginal lavage smear was collected to quantify the fungal load, and a Gram stain was utilized to assess the infection within each experimental group.

The day after the final *C. albicans* infection, all mice were anesthetized with 2% sodium pentobarbital (30 mg/kg), and vaginal tissues and blood samples were collected following a period of fasting. Certain mouse tissues were preserved in 4% paraformaldehyde solution for hematoxylin and eosin (H&E) staining, while others were stored at −80 °C for the assessment of mRNA transcription levels and inflammatory factor content.

### 2.17. Gram Staining of Mouse Vaginal Lavage Fluid

On the seventh day of *C. albicans* infection, the vaginas of mice were irrigated with PBS. Subsequently, 20 µL of the irrigated solution was collected on a slide, allowed to dry naturally, and then subjected to staining with crystal violet for 1 min, followed by iodine for 1 min. The slide was then decolorized with 95% ethanol for 30 sec and stained with a red solution for an additional 30 sec. The stained slide was allowed to dry naturally and subsequently examined under a microscope.

### 2.18. Fungal Load of Mouse Vaginal Lavage Fluid

On the seventh day of *C. albicans* infection, the vaginas of mice were irrigated with PBS, and 50 µL of the irrigated solution was coated onto YPD solid medium and incubated at 30 °C for 48 h. Subsequently, the fungal loads of each experimental group were visually assessed and quantified.

### 2.19. HE Staining of Mouse Vaginal Tissues

The vaginal tissues of each group were stained using the standard HE staining procedure, decolorized with varying concentrations of ethanol (70%, 80%, 90%) and xylene, and subsequently embedded in conventional paraffin wax. Transverse sections were then obtained to produce 4 μm tissue slices, which were affixed to slides. Subsequent examination under the microscope following H&E staining revealed tissue damage [34].

### 2.20. Enzyme-Linked Immunosorbent Assay

To investigate the ability of the strain to cause vaginal inflammation in mice. Quantitative analysis was conducted using a mouse tumor necrosis factor-α (TNF-α) enzyme-linked immunosorbent assay (ELISA) kit (Jiangsu Meiman Biotechnology Co., Ltd., Yancheng, China), mouse interleukin-6 (IL-6) ELISA kit (Jiangsu Meiman Biotechnology Co., Ltd., Yancheng, China), and mouse interleukin-10 (IL-10) ELISA kit (Jiangsu Meiman Biotechnology Co., Ltd., Yancheng, China) to quantify inflammatory factors.

The serum was obtained from mouse blood by centrifugation at 4 °C, 3000 rpm for 20 min, and subsequently analyzed using the ELISA kit protocol. The experimental procedure involved adding standards and samples to 96-well plates, incubating with enzyme reagent for 60 min, washing the plate with washing solution, adding color developer and termination solution, and measuring the absorbance (OD_450_ value) at 450 nm using an enzyme counter.

### 2.21. Statistical Analysis

The experiments were conducted with three biological replicates, and the data are presented as the mean ± standard deviation (SD). A Student’s *t*-test was applied for the comparison of two groups of data. Statistical analysis was performed using Prism 9.0 software (GraphPad Software, San Diego, CA, USA) with a significance level set at *p* < 0.05.

## 3. Results

### 3.1. OCT1 Is Involved in Maintaining Mitochondrial Homeostasis in C. albicans

To study the role of *OCT1* in mitochondrial functions, we employed the *HIS1-LEU2-ARG4* screening system to generate a single-gene deletion strain *oct1*∆::*HIS1*, the *oct1*∆/∆ mutant, and the *oct1* gene-reconstituted strain *oct1*∆/∆:*OCT1*. The validation results of polymerase chain reaction (PCR) and nutritional screening showed that the strains were successfully constructed (in Appendix A). Given the potential association between Oct1p and mitochondrial function, firstly, mitochondrial fluorescence dye Mito-Tracker Red was used to conduct mitochondrial staining on the *C. albicans* cells of each strain to observe the influence of Oct1p loss on mitochondrial morphology (Figure 1A). The results showed that the cells of WT contained 5–10 granular or rod-like mitochondrial tubules, which were discrete from each other. The *oct1*Δ/Δ cells contained several discrete granular mitochondrial tubules, most of which were dispersed in a fragmented state, and the *oct1*∆/∆:*OCT1* mitochondrial morphology was similar to the *oct1*Δ/Δ in the form of dispersed fragments. Then, we employed the BacTiter-Glo™ microbial cell activity detection kit to quantify intracellular adenosine triphosphate (ATP) levels in WT, *oct1*∆/∆, and *oct1*∆/∆:*OCT1* strains (Figure 1B). Specifically, the intracellular ATP concentration in WT cells was measured at approximately 738 nM, while the *oct1*∆/∆ mutant exhibited a markedly reduced intracellular ATP content of approximately 93 nM. The intracellular ATP content of the *oct1*∆/∆:*OCT1* strain measured approximately 407 nM, representing a significant increase in ATP levels compared to the *oct1*∆/∆ mutant. These findings indicate a crucial role for Oct1p in the regulation of intracellular ATP levels. Furthermore, the levels of mitochondrial membrane potential (MMP) were assessed using the JC-1 detection kit for the aforementioned three strains, providing insight into the aerobic capacity of cells (Figure 1C,D). The mean membrane potential ratio of WT was 2.6, while that of the *oct1*∆/∆ mutant was 1.4, a statistically significant decrease compared to the WT, indicating a potential role of Oct1p in modulating cellular membrane potential. The mean membrane potential ratio of the *oct1*∆/∆:*OCT1* strain was 1.6, demonstrating a substantial restoration relative to the *oct1*∆/∆ mutant. These findings support the hypothesis that the diminished membrane potential in the *oct1*∆/∆ mutant is attributable to the absence of Oct1p. Reactive oxygen species (ROS) is a by-product of cell growth and development. Excessive or insufficient production of ROS in cells can impact cell growth and development. In this study, intracellular levels of ROS were assessed using DCFH-DA probe staining (Figure 1E,F). The results showed that ROS production in the *oct1*∆/∆ mutant was significantly higher than that in WT, indicating that lack of Oct1p does lead to an abnormal increase in intracellular ROS production.

These results indicate the significance of Oct1p in cell growth and metabolism from multiple viewpoints, highlighting its potential role in regulating mitochondrial homeostasis.

### 3.2. Loss of Oct1p Affected the Growth of C. albicans

The carbon metabolism pathway of *C. albicans* primarily consists of glycolysis and aerobic respiration. Impairment of mitochondrial function is likely to impact the aerobic respiration process. Hence, to examine the impact of *OCT1* deficiency on the utilization of carbon sources and mitochondrial function in *C. albicans*, we employed glucose and glycerol-anhydrous ethanol blends as carbon sources in the medium. Subsequently, we assessed the growth conditions of WT, *oct1*∆/∆, and *oct1*∆/∆:*OCT1* strains at temperatures of 30 °C, 35 °C, and 37 °C, respectively (Figure 2A). There were no notable distinctions observed in the colony phenotypes of the WT, *oct1*∆/∆, and *oct1*∆/∆:*OCT1* strains when grown on glucose-enriched solid medium YPD. Conversely, when cultivated on glycerol-anhydrous ethanol-mixture-enriched solid medium YPEG, the *oct1*∆/∆ mutant displayed pronounced growth impairments relative to the WT. The exacerbation of these deficiencies was particularly pronounced at an elevated temperature of 37 °C, where the *oct1*∆/∆ mutant exhibited severely impaired growth. In contrast, the *oct1*∆/∆*:OCT1* strain displayed normal growth across all experimental conditions. Furthermore, the colony sizes of these strains were quantified and statistically analyzed using the coating culture method (Figure 2B). These findings indicate that the absence of Oct1p in *C. albicans* could potentially result in a reduction in the strain’s ability to metabolize glycerol anhydride ethanol. Additionally, the results suggest that Oct1p may have a significant impact on mitochondrial respiration, particularly as temperatures rise. The analysis of growth rates demonstrates that the growth of the *oct1*∆/∆ mutant is notably slower than that of the WT, indicating a clear positive regulatory role for Oct1p in the growth of *C. albicans* (Figure 2C). In addition, by analyzing the doubling time of each strain (Figure 2D), we found that the doubling time of WT was about 2 h, while that of *oct1*Δ/Δ was about 2.6 h, which was significantly increased compared with WT, and that of *oct1*∆/∆:*OCT1* was about 2.2 h, which recovered significantly compared with *oct1*Δ/Δ. Prior research has indicated that disruptions in mitochondrial function in *C. albicans* are frequently linked to changes in cell wall integrity or membrane function. In light of this, we assessed the susceptibility of the WT, *oct1*∆/∆ mutant, and *oct1*∆/∆:*OCT1* strain to stress-inducing agents that target the cell membrane and cell wall. Specifically, we exposed the strains to Congo Red (CR) and Calcofluor White (CFW) in the growth medium to induce cell wall stress. In relation to membrane stress, FLU and AmB were chosen as antifungal reagents alongside SDS. The results indicate that the *oct1*∆/∆ mutant exhibited significant resistance to fluconazole and SDS, while displaying no discernible abnormal phenotypes in response to cell wall pressure agents CFW, CR, or AmB (Figure 2D and Appendix A).

### 3.3. Transcriptomic Analysis of the OCT1 Functions

To further investigate the role of Oct1p in *C. albicans*, transcriptome sequencing was conducted on the *oct1*∆/∆ mutant on three biological replicates, followed by a comparative analysis with the wild-type strain. The difference in expression levels was determined based on the criteria of a |log2fold change| > 1, with a false discovery rate of 0.05, for the identification of differentially expressed genes (DEGs) (Figure 3A). In comparison to the WT, the *oct1*∆/∆ mutant exhibited an increased expression of 614 genes and a decreased expression of 239 genes. Furthermore, gene ontology (GO) functional enrichment analysis was performed on these differentially expressed genes to investigate the involvement of OCT1 in *C. albicans*. Our analysis revealed that the down-regulated genes were significantly enriched in functions such as electron carrier activity (36.3%), cell aggregation (8.3%), transporter activity (6.2%), biological adhesion (3.5%), single-organism process (3.2%), and multi-organism process (3.1%). Conversely, the up-regulated genes exhibited enrichment in immune system processes (50%), structural molecule activity (30.9%), organelle part (9.1%), electron carrier activity (9.1%), macromolecular complex (7%), and multicellular organismal processes (6.7%) (Figure 3B,C). Significantly, the *oct1*∆/∆ mutant exhibited decreased expression levels of genes linked to cell aggregation and biological adhesion, suggesting a potential impact of Oct1 loss on the adhesion and aggregation capabilities of *C. albicans*. These processes play a crucial role during hyphal growth. Subsequently, we conducted a screening of annotated genes associated with *C. albicans* adherence, growth, and aggregation within the differentially expressed genes (DEGs) of the *oct1*∆/∆ mutant, utilizing the Kyoto Encyclopedia of Genes and Genomes (KEGG) database for enrichment analysis. The expression levels of seven genes were identified (Figure 3D), including genes such as *ECE1*, which has been associated with the virulence factors of *C. albicans*, such as adhesion, biofilm formation, and filamentation properties, and *ALS1* and *ALS3* in the lectin-like sequence family that plays essential roles in the processes of adherence and biofilm formation in vitro, as well as the high-affinity iron permease *FTR1*, a gene crucial for *C. albicans* growth [35,36,37]. To further confirm the transcription levels of these genes, quantitative real-time PCR was used to verify the expression levels of these four genes. The expression levels of these four genes were down-regulated in the *oct1*∆/∆ mutant compared with the WT, which was consistent with the results in the transcriptome (Figure 3E). These results suggest that *OCT1* may be involved in the adhesion, aggregation, and hyphal growth of *C. albicans*.

### 3.4. Loss of Oct1p Resulted in the Impaired Hyphal Growth and Biofilm Formation of C. albicans

To further confirm the functions of Oct1p, we examined the abilities of the hyphal formation, biofilm formation, and adhesion of the corresponding mutants, respectively. First, we observed and analyzed the hyphal formation status of the WT, the *oct1*∆/∆ mutant, and the *oct1*∆/∆:*OCT1* that were cultured in six different liquid media: RPMI-1640, YPD, YPD + 10% FBS, Lee’s, Spider, and SLAD (Figure 4A). The WT exhibited normal germination and hyphal growth across all culture conditions, whereas the *oct1*∆/∆ mutant displayed severely restricted hyphal growth, failing to reach the levels observed in the WT and forming only short pseudohyphae. The hyphal morphology of the *oct1*∆/∆:*OCT1* strain closely resembled that of the WT. Furthermore, we conducted a comparative analysis of the colony growth among these three strains on five distinct solid media: YPD, YPD + 10% FBS, Lee’s, Spider, and SLAD (Figure 4B). In comparison to the WT, the *oct1*∆/∆ mutant exhibited a notable reduction in colony size with predominantly smooth edges and an inability to form normal hyphae. Conversely, the *oct1*∆/∆:*OCT1* mutant displayed a restoration of solid hyphal formation capability when compared to the *oct1*∆/∆ mutant. These findings suggest that Oct1p plays a crucial role in the hyphal growth process of *C. albicans*. Then, we explored the effect of Oct1p on the biofilm formation of *C. albicans* by XTT (dimethoxazole yellow) detection and CFW staining (Figure 4C–E). The results showed that the biofilm formation of the *oct1*∆/∆ mutant was significantly reduced compared with the WT. Finally, through the detection of the adhesion of the WT, the *oct1*∆/∆ mutant, and the *oct1*∆/∆:*OCT1* (Figure 4F), we found that the adhesion of the *oct1*∆/∆ mutant was also significantly weakened compared with the WT. These results suggest that OCT1 plays an important role in hyphal growth, adhesion, and biofilm formation.

### 3.5. Deletion of OCT1 Led to the Decreased Virulence of C. albicans

Numerous studies have demonstrated that biofilm formation is crucial for the pathogenesis and virulence of *C. albicans*. To assess the influence of *OCT1* on virulence, an experiment was conducted using the larvae of the invertebrate model, *G. mellonella*. Larvae were infected with cells from three strains (WT, *oct1*∆/∆ mutant, and *oct1*∆/∆:*OCT1*), and their survival rates were monitored (Figure 5A). Results showed that all the larvae infected by the WT died on day 6, while about half of the larvae infected by the *oct1*∆/∆ mutant survived by the end of the day. The survival rate of larvae infected by the *oct1*∆/∆ mutant was significantly lower than that of the WT. All the larvae infected by *oct1*∆/∆:*OCT1* died on day 9. In addition, the fungal load in the *G. mellonella* was determined 48 h after infection (Figure 5B). The results showed that the fungal load in the larvae of the *oct1*∆/∆ mutant was significantly decreased compared to that of the WT-infected larvae. Meanwhile, the fungal load in *oct1*∆/∆:*OCT1*-infected larvae was significantly increased compared to that of the *oct1*∆/∆ mutant.

Subsequently, we assessed the impact of Oct1 on the pathogenicity of *C. albicans* by inducing vulvovaginal infection in mice. Specifically, we inoculated the vaginas of mice with WT, *oct1*∆/∆, and *oct1*∆/∆:*OCT1* strains of *C. albicans*. Following this, we conducted observations of the vaginal orifice, vagina, and vaginal douching fluid of the mice at a 7-day post-infection time point (Figure 5C). The vulvar vaginal orifice of mice infected with the WT exhibited conspicuous signs of inflammation, including redness, swelling, and ulceration. Conversely, mice infected with the *oct1*∆/∆ strain displayed a relatively normal appearance in the vulvar vaginal orifice. However, mice infected with the rescued strain *oct1*∆/∆:*OCT1* showed noticeable redness and swelling in the vulvar vaginal orifice. The examination of the mice’s vaginal tissue using hematoxylin and eosin (HE) staining revealed that the vaginal mucosa of mice in the control group exhibited an intact structure with a fully developed keratinized layer on the surface. Conversely, in the WT-infected WT group, the keratinized layer on the mucosal surface was absent, and there was notable evidence of squamous epithelial cell proliferation, accompanied by a substantial infiltration of inflammatory cells into the mucosa. Conversely, in the *oct1*∆/∆ group, there was a presence of a certain amount of keratinized layer on the surface, and a reduction in inflammatory cell infiltration was noted. The extent of damage to the vaginal tissue in the *oct1*∆/∆:*OCT1* group was more severe compared to the *oct1*∆/∆ group.

When we performed Gram staining on the vaginal irrigation fluid, we observed that *C. albicans* in the *oct1*∆/∆ group exhibited minimal or no hyphal growth and did not adhere to the vaginal epithelial cells. In contrast, a significant number of slender hyphae grew and adhered to the vaginal epithelial cells, forming clusters in both the WT group and the *oct1*∆/∆:*OCT1* group. Furthermore, a vaginal irrigation solution was spread onto the plate to determine the fungal load in the vaginal cavities of mice (Figure 5D). It was observed that the fungal load in the *oct1*∆/∆ group exhibited a notable decrease in comparison to the WT group, whereas in the *oct1*∆/∆:*OCT1* group, there was a significant increase observed when compared to the *oct1*∆/∆ group. At the same time, an enzyme-linked immunosorbent assay (ELISA) was used to detect changes in the levels of two pro-inflammatory factors (IL-6, TNF-α) and one anti-inflammatory factor (IL-10) in mouse serum. The results (Figure 5E) indicated that compared to the blank control group the levels of pro-inflammatory factors in the serum of the WT group were significantly increased, while the levels of the anti-inflammatory factor were significantly decreased. Compared with the WT group, the *oct1*∆/∆ group showed a significant reduction in the content of pro-inflammatory factors and a significant increase in anti-inflammatory factors. Conversely, the *oct1*∆/∆:*OCT1* group exhibited a significant increase in pro-inflammatory factors and a decrease in anti-inflammatory factors compared to the WT group. This result suggests that the loss of Oct1 does lead to a significant reduction in the virulence of *C. albicans*. In addition, we also measured the transcription levels of three inflammatory factors in each group using qRT-PCR, and the results were largely consistent with the ELISA results (Figure 5F). Compared with the blank control group, the expression of pro-inflammatory factors in the serum of the WT group was significantly increased, while the expression of anti-inflammatory factors was significantly decreased. Compared with the WT group, the *oct1*∆/∆ group showed a significant reduction in the expression of pro-inflammatory factors and a significant increase in anti-inflammatory factors. In contrast, the *oct1*∆/∆:*OCT1* group exhibited a significant increase in pro-inflammatory factors compared to the *oct1*∆/∆ group, while the expression of anti-inflammatory factors was significantly decreased. These results suggest that the absence of the Oct1 diminishes *C. albicans*’ ability to infect the host.

## 4. Discussion

In this research, we selected the mitochondrial protease Oct1p in *C. albicans* as the focal point of our investigation. Its counterpart in *S. cerevisiae* has been identified as an octapeptide aminopeptidase localized within the mitochondrial matrix, which plays an important role in the processing of mitochondrial proteins [15]. In this study, we constructed the *oct1*∆/∆ mutant by homologous recombination, as well as the *OCT1* gene-reconstituted strain *oct1*∆/∆:*OCT1*. A series of studies was undertaken to investigate their phenotype. Initially, our investigation revealed a significant decrease in intracellular ATP content and mitochondrial membrane potential in the *oct1*∆/∆ mutant compared to the WT. Additionally, we observed a marked increase in reactive oxygen species levels in the cytoplasm of the *oct1*∆/∆ mutant relative to the wild type. All three parameters are crucial indicators of typical mitochondrial function, with ATP serving as a direct product of mitochondrial aerobic respiration. The mitochondrial membrane potential provides a visual representation of the electron transfer process within the mitochondrial respiratory chain [38]. While ROS are commonly understood as by-products of respiration, the oxidative damage caused by an excess of ROS typically indicates mitochondrial dysfunction [38]. The simultaneous dysregulation of all three factors in the *oct1*∆/∆ mutant implies a significant role of the Oct1p in the mitochondrial processes of *C. albicans*. Furthermore, it is important to note that compromised mitochondrial function could potentially disrupt aerobic respiration, a crucial component of the carbon metabolic pathway [39,40]. Thus, we conducted an investigation into the growth status of the *oct1*∆/∆ mutant in media supplemented with various carbon sources. Our findings revealed that the absence of Oct1p not only hindered the growth of the strain in glucose-containing media but also exacerbated the growth impairment in glycerol-containing media. The exacerbation of growth defects with elevated temperatures suggests a potential role for Oct1p in the aerobic respiration pathway of *C. albicans*. Furthermore, to identify any anomalies in the cell wall or cell membrane functionality of the *oct1*∆/∆ mutant, various stress reagents were introduced into the culture medium, and the growth of the strain was monitored. Ultimately, it was observed that the *oct1*∆/∆ mutant displayed an increased tolerance to fluconazole and SDS compared to the WT. Given that azoles exert their inhibitory effects by impeding the synthesis of fungal ergosterol, it is hypothesized that the deletion of *OCT1* may have resulted in the aberrant expression of the gene responsible for ergosterol production, consequently influencing the resistance of *C. albicans*. In order to validate this hypothesis, RNA-seq analysis was conducted, revealing a significant up-regulation of the *ERG11* gene expression in the *oct1*∆/∆ mutant when compared to the WT (Appendix A). The *ERG11* gene, which encodes cytochrome P450 lanosterol 14α-demethylase (Erg11p), is a primary target of azole antifungal drugs [41]. We propose that the lack of the Oct1p leads to elevated *ERG11* gene expression, causing an overproduction of ergosterol and consequently leading to the development of azole resistance in *C. albicans* [42]. Regarding the tolerance of the *oct1*∆/∆ mutant to SDS, it is hypothesized that this phenomenon may be attributed to a modification in ergosterol levels, which could alter the structure of its plasma membrane.

The results of GO enrichment analyses indicated that the lack of Oct1p resulted in alterations in gene expression within crucial pathways, such as cell aggregation, bioadhesion, and hyphal formation. We found a significant down-regulation in the transcript levels of four regulators associated with hyphal growth and adhesion in the *oct1*∆/∆ mutant compared to the WT. These regulators consist of *ECE1*, which has been linked to the adhesion, biofilm formation, and filamentation characteristics of *C. albicans*; lectin-like sequence families *ALS1* and *ALS3* involved in in vitro biofilm formation; and *FTR1*, a gene essential for the growth of *C. albicans* during systemic infection [35,37,43]. Based on the results of RNA-seq analysis, we further observed the effect of *OCT1* deletion on hyphal formation. Firstly, the hyphal formation ability of the WT and the *oct1*∆/∆ mutant was compared in liquid and solid hyphal formation media, respectively. It was observed that the *oct1*∆/∆ mutant was unable to form normal hyphae. In addition to the abnormal process of hyphae formation, we also observed a significant reduction in the biofilm formation ability and adhesion ability of the *oct1*∆/∆ mutant compared to the WT. Therefore, Oct1p plays a crucial role in the morphological transformation, hyphal formation, adhesion, and biofilm formation of *C. albicans*.

Despite evidence suggesting that dysfunctional mitochondria in *C. albicans* can hinder the formation of hyphae [6], the precise mechanisms governing morphological transitions and the regulation of hyphal growth in response to mitochondrial function in this organism are not yet fully understood. Furthermore, research has demonstrated that the transition of *C. albicans* from the yeast state to the hypha state is intricately linked to its pathogenicity [44]. Fungal cell virulence is often determined by their ability to adhere and form biofilms. This is supported by the observation that *C. albicans* cells extracted from mature biofilms exhibit enhanced adhesion capabilities and biofilm-forming abilities [45]. Based on the observed phenotypes of the *oct1*∆/∆ mutant, it is hypothesized that the absence of Oct1p may influence the virulence of *C. albicans*. To investigate this hypothesis, two infection models involving *G. mellonella* and the mouse vagina were utilized to assess the virulence of the *oct1*∆/∆ mutant. Ultimately, the results indicate that the *oct1*∆/∆ mutant exhibits significantly reduced virulence compared to the WT. Furthermore, through staining, microscopic observation, and detection of fungal load in the vaginal lavage fluid of mice, our study revealed a significantly lower residual amount of hyphae in the vaginal lavage fluid of mice infected with the *oct1*∆/∆ mutant compared to the WT. Additionally, the *oct1*∆/∆ group exhibited predominantly sprouting spores or very short pseudohyphae, with a notable absence of mature hyphae. This suggests that the reduced virulence of the *oct1*∆/∆ mutant in *C. albicans* may be attributed to its impaired ability to form hyphae, a characteristic closely linked to virulence. Additionally, analysis of immune factor expression in the vaginal tissues of infected mice revealed that the *oct1*∆/∆ mutant elicited a weaker immune response compared to the wild-type strain, further supporting the notion that the knockout of *OCT1* resulted in a less virulent phenotype. In summary, the findings of our study indicate that Oct1p is involved in the regulation of cellular, hyphal, and biofilm growth processes in *C. albicans* by affecting mitochondrial function, as well as in interactions with the host that contribute to the formation of pathogenicity (Figure 6).

The current study highlights the crucial role of Oct1p, a key mitochondrial protease, in regulating mitochondrial homeostasis in *C. albicans*. Depletion of Oct1p results in significant disruptions in hyphal formation, adhesion, and biofilm development in *C. albicans*, ultimately reducing its virulence. This research demonstrates the significance of mitochondria in facilitating *C. albicans* hyphal growth and virulence from a focused perspective. It provides a strong theoretical basis for addressing Candida-related illnesses and advancing the development of new therapeutic agents. Furthermore, it provides a valuable point of reference for investigating the relationship between mitochondria and virulence in other pathogenic fungi.

## Figures and Tables

**Figure 1 jof-10-00391-f001:**
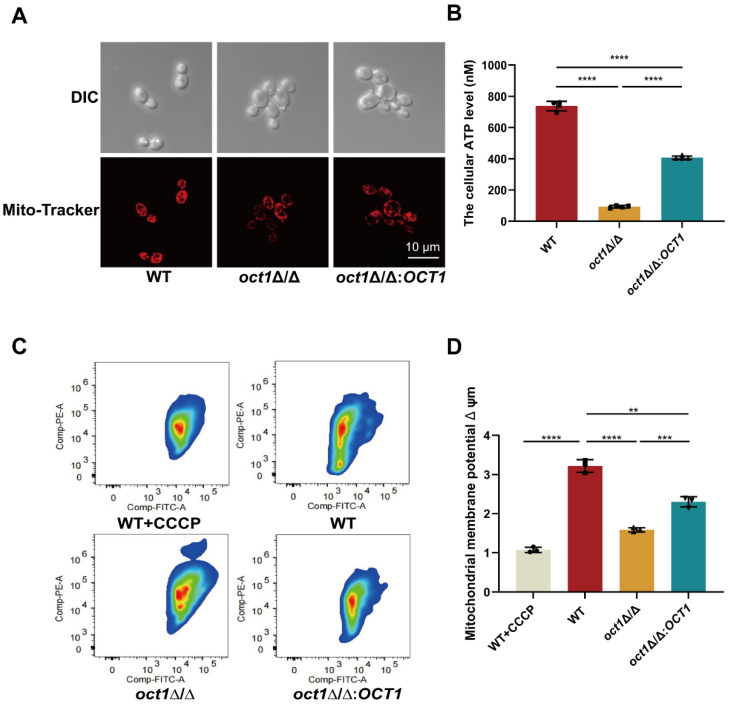
Deletion of *OCT1* reduces *C. albicans* mitochondrial function. (**A**) Mitochondrial morphology under 579 nm excitation was observed. The concentration of *C. albicans* suspension was adjusted with PBS to 1.0 × 10^6^ cells/mL for each strain, and then 200 μM Mito-tracker Red dye solution was added to the suspension at the ratio of 1:1000 and incubated at 30℃ in the dark for 30 min. Scale bar: 2 μm. (**B**) A total of 100 μL of 1.0 × 10^7^ cells/mL of each *C. albicans* liquid and an equal volume of BacTiter-Glo^TM^ reaction solution were added to an opaque 96-well plate and incubated for 5 min at room temperature after oscillating and mixing, and the ATP content of each sample was detected by a multifunctional enzyme marker. (**C**) Mitochondrial membrane potential (MMP) was assessed using the JC-1 assay kit and flow cytometry (Ex/Em of 595/488 nm). (**D**) The average red/green fluorescence intensity was recorded for each sample. The transition of JC-1 dye from red to green fluorescence was used to easily detect the decrease in MMP, after which MMP was determined as the ratio of red to green fluorescence. (**E**) ROS levels of WT, *oct1*∆/∆, and *oct1*∆/∆:*OCT1* were detected using a DCFH-DA probe and visualized using a microscope (Ex/Em of 488/525 nm). Scale bar: 20 μm. (**F**) The fluorograms of each strain were quantitatively analyzed using Image J software. (ns, not significant; **, *p* < 0.01; ***, *p* < 0.001; and ****, *p* < 0.0001).

**Figure 2 jof-10-00391-f002:**
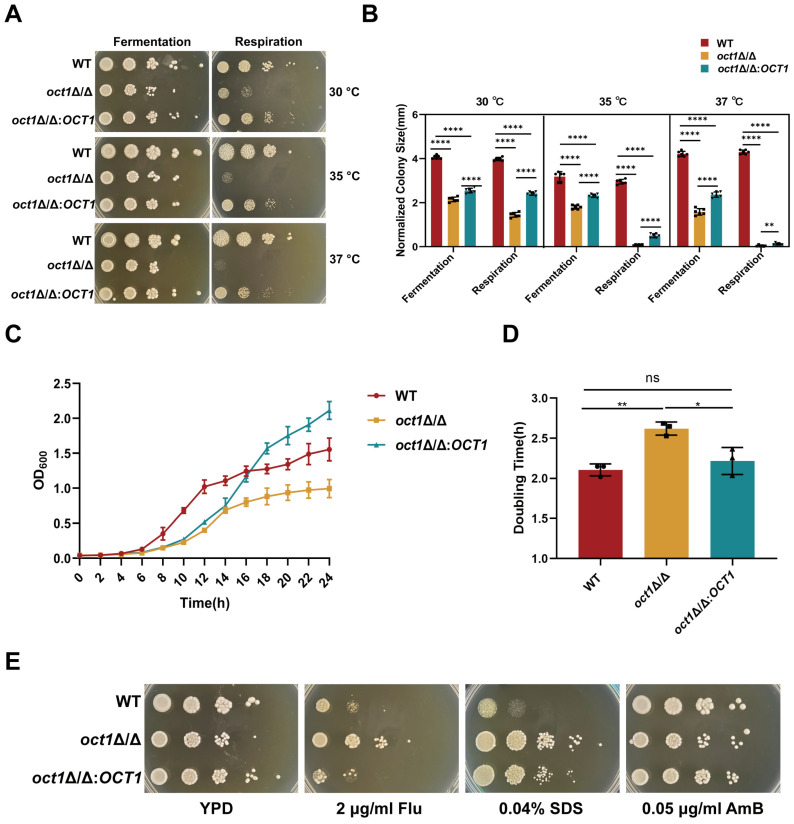
Oct1p affects *C. albicans* growth ability. (**A**) Each strain was incubated overnight in YPD, then washed with PBS and serially diluted, and different concentrations of WT, *oct1*∆/∆, and *oct1*∆/∆:*OCT1* (5 μL 10^6^ cells/mL to 10^2^ cells/mL) were spotted on YPD solid medium containing 2% dextrose as a carbon source and 3% glycerol and anhydrous ethanol mixture as a carbon source and YPEG solid medium with 3% glycerol and anhydrous ethanol mixture as a carbon source. The plates were then incubated at 30 °C, 35 °C, and 37 °C for 2 days and then photographed. (**B**) Each strain was incubated in YPD overnight, then diluted to 500 cells/mL with sterile PBS, and 100 μL was spread on YPD solid medium containing 2% glucose as a carbon source and YPEG solid medium with a 3% mixture of glycerol and anhydrous ethanol as a carbon source, respectively, and then incubated for 2 days at 30 °C, 35 °C, and 37 °C, respectively, and then measured by using a vernier caliper, the size of single colonies, and recorded. (**C**) Each strain was incubated at 200 mL with an initial concentration of OD_600_ = 0.04 on a shaker at 30 °C, 200 rpm for 24 h, and OD_600_ was measured at 2 h intervals; the data were recorded, and growth curves were plotted. (**D**) Doubling time of *C. albicans* in the exponential growth phase. (**E**) Each strain was incubated overnight in YPD, then washed with sterile PBS and serially diluted. Different concentrations of WT, *oct1*∆/∆, and *oct1*∆/∆:*OCT1* (5 μL 10^6^ cells/mL to 10^2^ cells/mL) were spotted on YPD, fluconazole containing 2 μg/mL, 0.04% SDS, and amphotericin B medium at 0.05 μg/mL, and the plates were photographed after incubation at 30 °C for 2 days. (ns, not significant; *, *p* < 0.05; **, *p* < 0.01; ****, *p* < 0.0001).

**Figure 3 jof-10-00391-f003:**
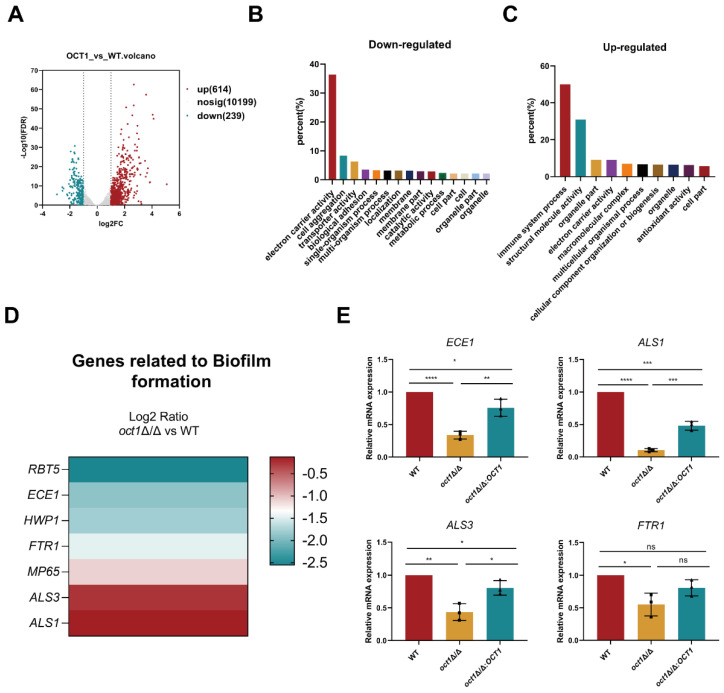
RNA-seq analysis reveals the potential functions of Oct1p. (**A**) Genes (DEGs) significantly differentially expressed in *oct1*∆/∆. Expression difference multiples |log2fold change| > 1 and false discovery rate *p*-value < 0.05. (**B**) Major distribution of *oct1*∆/∆ down-regulated differential genes in GO function enrichment analysis (no distinction between molecular function, cellular components, and biological processes). (**C**) Primary distribution of *oct1*∆/∆ up-regulated differential genes in GO functional enrichment analyses (no distinction between molecular function, cellular components, and biological processes). (**D**) Heat map of up- and down-regulated genes associated with biofilm formation of the KEGG category. (**E**) Transcript levels of genes associated with *C. albicans* hyphal growth, adhesion in WT, and *oct1*∆/∆ strains were verified by qRT-PCR in genome-wide transcriptional data. (ns, not significant; *, *p* < 0.05; **, *p* < 0.01; ***, *p* < 0.001; and ****, *p* < 0.0001).

**Figure 4 jof-10-00391-f004:**
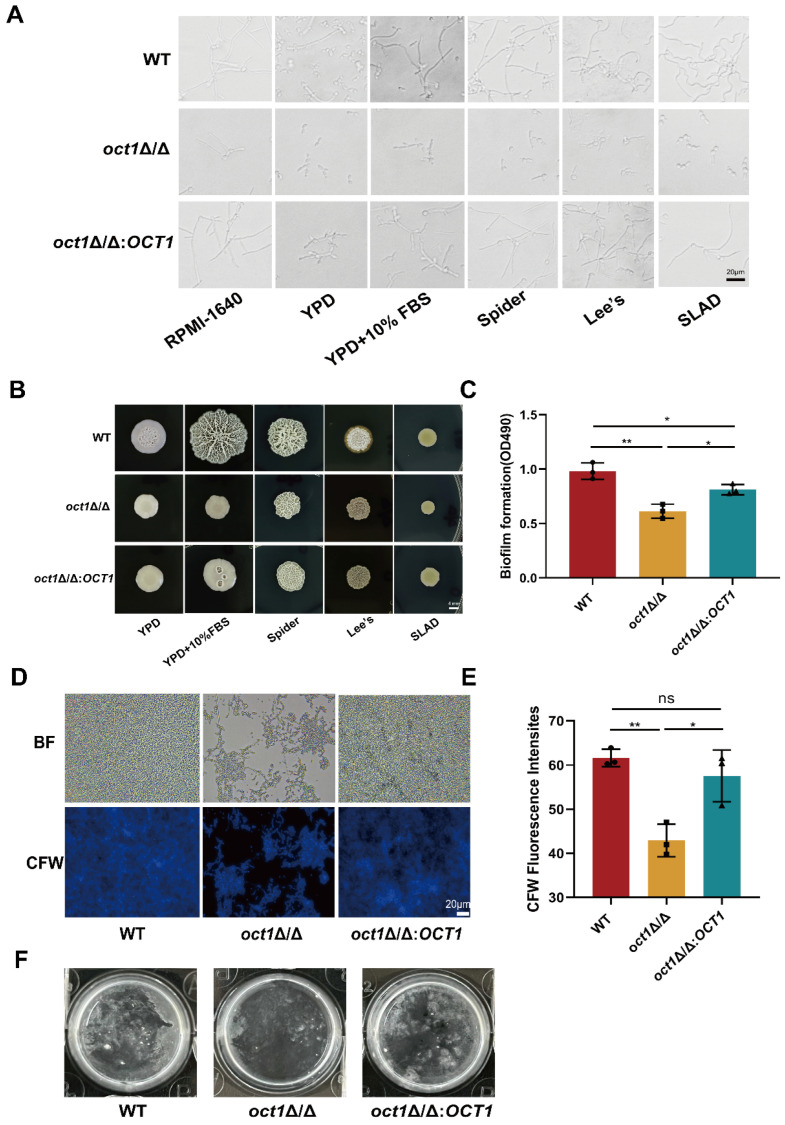
Deficiency of Oct1p attenuates *C. albicans* hyphal formation, biofilm formation, and adhesion. (**A**) Each strain was incubated in YPD overnight, then washed twice with sterile PBS, and incubated in six different liquid mycelium induction media: RPMI-1640, YPD, YPD + 10% FBS, Lee’s, Spider, and SLAD. The concentration of each solution was adjusted to 1.0 × 10^6^ cells/mL, and 2 mL of each solution was added into 6-well plates at 37 °C for 4 h. The formation of liquid mycelium was observed under a microscope. Scale bar: 4 μm. (**B**) Each strain was incubated in YPD overnight, and then the concentration of each strain was adjusted to 1.0 × 10^5^ cells/mL after being washed twice with sterile PBS, and 5 μL was spotted on five different solid mycelium-inducing media (YPD, YPD + 10% FBS, Lee’s, Spider, and SLAD) and incubated in a 37 °C incubator. The results were observed and photographed for recording. Scale bar: 4 mm. (**C**) XTT reduction assay to detect the in vitro biofilm formation ability of WT, *oct1*∆/∆, and *oct1*∆/∆:*OCT1*. (**D**) Each strain was cultured in YPD overnight, then washed twice with sterile PBS, and then the concentration of each bacterial solution was adjusted to 1.0 × 10^6^ cells/mL with RPMI-1640 liquid medium, incubated at 37 °C for 24 h, and then observed under a fluorescence microscope. After incubation at 37 °C for 24 h, the *C. albicans* were stained with Calcium fluorescent white for 5 min and then observed under a fluorescence microscope. Scale bar: 40 μm. (**E**) The fluorograms of each strain were quantitatively analyzed using Image J software. (**F**) WT, *oct1*∆/∆, and *oct1*∆/*OCT1* in vitro adhesion capacity assay. (ns, not significant; *, *p* < 0.05; **, *p* < 0.01).

**Figure 5 jof-10-00391-f005:**
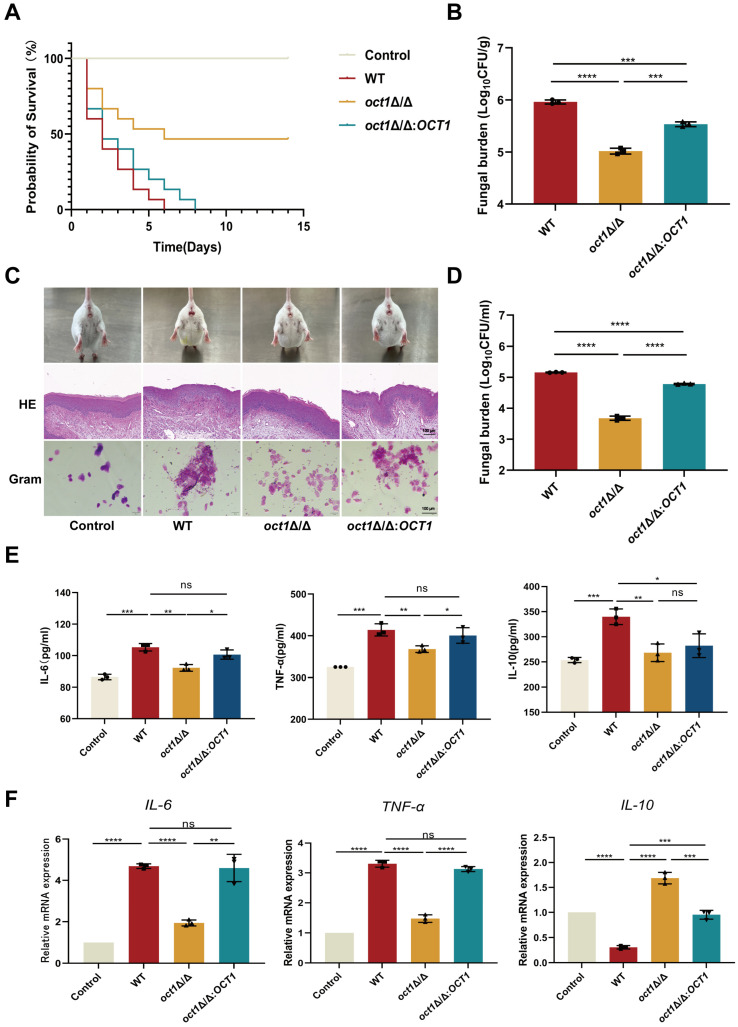
The absence of Oct1p reduces the infective ability of *C. albicans* in *G. mellonella* and mice. (**A**) The concentration of each *C. albicans* liquid was adjusted to 8.0 × 10^6^ cells/mL with PBS, and 10 μL of each *C. albicans* liquid was introduced and injected into the left or right hypophysis of *G. mellonella* larvae using a pointed microinjection needle, and then placed in the incubator at 37 °C after making the molds and incubated, and then the deaths of *G. mellonella* were observed and recorded every day. The experiment lasted for 14 days. (**B**) Fungal load in the *G. mellonella* at 48 h after the injection of each fungal solution into the larvae of the *G. mellonella*. (**C**) Localized state of the mouse vagina, histopathological changes in the vaginal mucosa, and the morphology of *C. albicans* in vaginal lavage fluid 7 days after infection with vulvovaginal candidiasis. Scale bar: 100 μm. (**D**) Fungal load in the vaginal lavage fluid of mice 7 days after infection with vulvovaginal candidiasis. (**E**) The method of ELISA was used to detect the levels of IL-6, TNF-α, and IL-10 in mouse vaginal tissues. (**F**) The method of qRT-PCR was used to detect the expression of *IL-6*, *TNF-α*, and *IL-10* genes in mouse vaginal tissues. (ns, not significant; *, *p* < 0.05; **, *p* < 0.01; ***, *p* < 0.001; and ****, *p* < 0.0001).

**Figure 6 jof-10-00391-f006:**
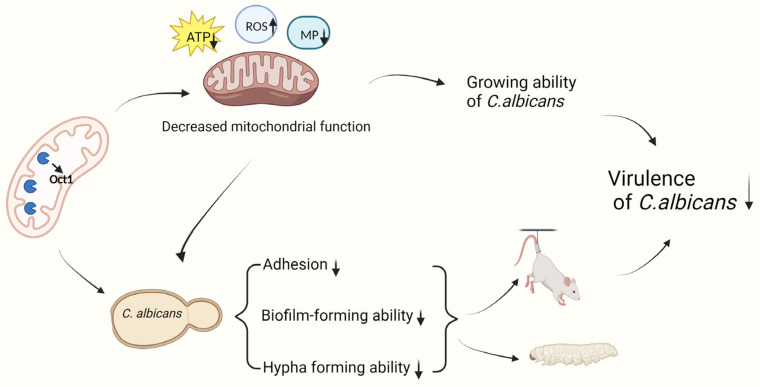
Diagram of the working mode of Oct1p in *C. albicans*. The upward or downward arrow indicates an increase or decrease in the corresponding parameter.

## Data Availability

Data are contained within the article and Appendix A.

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
