# Peer review of "Mitochondrial Protease Oct1p Regulates Mitochondrial Homeostasis and Influences Pathogenicity through Affecting Hyphal Growth and Biofilm Formation Activities in Candida albicans"

_jof, 2024, doi:10.3390/jof10060391_

Round 1

Reviewer 1 Report

The article by Zhu et al describes intesively the role of OCT1 in regulating cellular morphology, virulence, and drug resistance mechanisms. The article is very well written supported by extensive data sets. Here are my comments:

1) The introduction will greatly benefit if the authors can provide what is known about the role of mitochondrial function in regulating cellular morphology, virulence, and drug resistance in pathogenic fungi other than C. albicans as well. Please explain.

2) Through out the text OCT1 needs to be italicized or proper protein nomenclature for C. albicans should be used to designate the protein wherever the authors have mentioned it. Please clarify.

3)Did the authors consider looking at the structure of the mitochondria in the mutant OCT1 cells. Since, mitochondrial structure can affect it functions.

4)Fig 3C: The authors may consider analysis doubling times of the curve. The revertant OCT1 's growth seems to be affected in the same way as the mutant and not in the same way as the wild type. Please clarify.

5)Fig 7: Please explain the figure in the discussion section for better clarity.

6)Fig 4: Was RNAseq performed in biological duplicate or triplicate. Please clarify.

7)Fig 4: Since qPCR was used to verify, please describe quantity of RNA used in the experiment in materials and methods. What was the housekeeping gene used as control for the qPCR?

Same as above.

Reviewer 2 Report

This work presents a comprehensive phenotypic characterization of the OCT1 mutants of Candida albicans, a major fungal pathogen of humans. OCT1 encodes a putative metalloendopeptidase located in the mitochondrial matrix. In this study, the authors constructed a gene deletion mutant and a rescued strain. They then investigated the phenotypes of the mutant relevant to mitochondrial functions, such as ATP production, mitochondrial membrane potential, ROS levels, and carbon source utilization, confirming that the loss of OCT1 causes mitochondrial dysfunction. They further demonstrated that the oct1 mutant is defective in hyphal growth, biofilm formation, and adhesion. Finally, using the Galleria mellonella larvae and mouse vulvovaginal candidiasis models, they demonstrated the attenuated virulence of the OCT1 mutant.

To my knowledge, this is the first study that characterizes the OCT1 mutants. It covered a wide ground and produced interesting data regarding the role of OCT1 in hyphal growth, adhesion, and biofilm formation, as well as in mitochondrial functions. However, the manuscript was not well written and is difficult to read due to many language issues. The authors should seek help from individuals proficient in English to polish it.

The author should address the following issues:

Major issues

(1)    The phylogenetic analysis data shown in Figure 1 is not necessary, which can be removed. This manuscript is already quite long, containing many experiments. Removing the data does not affect the presentation of this study.

(2)    Lines 341-349. The results of these experiments are not sufficient to confirm the success of the mutant construction. You need to demonstrate that both alleles of the OCT1 gene have been correctly replaced by the selectable marker genes by performing PCR, using primers flanking the target gene. The data shown in Figure 1A only shows that the selectable marker genes have been introduced into the genome but do not show that they were integrated in the right locus.

(3)    Figure 4E. The significantly higher expression levels of ECE1, ALS1 and ALS3 in the rescued strain are very strange. How do you explain why the re-introduction of a single copy of OCT1 can result in a level of expression twice as high as in the WT cells, which have two copies of OCT1? In contrast, this strain shows only partial restoration in the levels of ATP, mitochondrial membrane potential, and ROS as shown in Figure 2B-F. Something must be wrong in the RT-qPCR experiments.

(4)    Lines 463-464. The description ‘ECE1, which is a key gene affecting hyphal production ability’ is incorrect. ECE1 encodes a peptide toxin specifically expressed during hyphal growth, but it is not required for hyphal growth. Despite being hyphae-specific, ASL1 and ALS3 are also not required for hyphal growth. The description ‘FTR1, a gene crucial for C. albicans growth’ is in accurate. FTR1 encodes a high-affinity iron transporter that is required for growth during systemic infection. When you make these statements, you should cite the relevant publications.

Minor issues

Lines 33-34. The sentence can be shortened to “Infections caused by C. albicans range from superficial conditions to systemic diseases.

Line 36. the body’s microflora

Line 57. Regarding the nomenclature of gene and protein names, gene names should be written in italiic capital letters, such as OCT1, and protein names as Oct1 or Oct1p. Should italicize all OCT1 throughout the manuscript.

Line 70. reduces

Line 84. Please check Alistair J.P. Brown’s address. He was at the University of Aberdeen previously and now is at the University of Exeter.

Line 96. By convention the rescued strain should be named as oct1∆/∆:OCT1. Oct1∆/OCT1 is for the heterozygous mutant. Please change throughout the manuscript.

Line 126. Please change ‘C. albicans solution’ to ‘C. albicans culture or suspension’ throughout the manuscript. It is wrong to use ‘solution’ here, as it typically refers to a homogeneous mixture of one or more substances dissolved in a solvent, often used in the context of chemistry.

Line 127. add a period after PBS.

Line 132. the impact of deleting OCT1. For genes, you use deletion; for protein you use depletion.

Line 142. Capitalize the first letter: Mitochondrial Membrane Potential Assay

Line 140. it reads better to use ‘in the dark’ to replace ‘in the absence of light’.

Line 151. the impact of OCT1 deletion

Lines 164-165, Rephrase the sentence. containing different carbon sources and stress reagents, and then incubated at various temperatures for 48 h to observe growth.

Line 165. delete the word ‘this’.

Line 182. Shouldn’t it be stationary phase instead of exponential phase?

Line 184. at 30 °C with shaking at 200 rpm for 24 h

Line 189. I believe it should be: The overnight culture was inoculated into fresh YPD for further cultivation, and the cells were harvested at the log phase for cryopreservation in liquid nitrogen.

Line 194. Delete ‘growth at’ so it reads ‘until the log phase’. At the end of the line, ‘use’ not ‘using’.

Line 214. The development of hyphae was monitored. Delete the word ‘solid’.

Line 218-219. hyphal growth was examined. Delete the word ‘liquid’.

Line 227. It should be ‘the biofilm was washed twice with PBS.

Lines 235-236. the deletion of OCT1

Line 258The impact of OCT1 deletion

Line 262. Replace the word ‘principles’ with ‘regulations’.

Line 338. To study the role of OCT1 in mitochondrial functions

Line 417. Condo Red (CR) and Calcofluor White (CFW) in the growth medium

Line 433. In 200 mL of medium

Line 434. OD600 was measured

Line 442. Replace ‘The discrepancy’ with ‘Difference’.

Line 458. Rephrase the sentence. These processes play a crucial role during hyphal growth.

Line 525. Deletion of OCT1

Line 530. Results show

Line 534. Replace the word ‘detected’ with ‘determined’.

Line 537. Delete ‘the larvae of’; add a hyphen after OCT1. Fungal load in oct1∆/∆:OCT1-infected

Lines 546-547. With the rescued strain oct1∆/∆:OCT1

Line 551. Replace the word ‘model’ with wild-type or WT.

Line 560. Replace the word ‘model’ with wild-type or WT.

Lines 561-562. was spread onto the plate to determine the fungal load in the vaginal cavities of mice

Line 574. Replace the word ‘model’ with wild-type or WT.

Line 586. infect the host.

In Figure 6. Change the label ‘Model’ to ‘WT’.

Line 618. Replace ‘byproduct’ with ‘product’.

Round 2

Reviewer 2 Report

The authors have addressed the issues I raised satisfactorily.

I suggest citing the Science paper published by R Narendrakumar et al. that first reported the discovery of FTR1 to replace Ref. 36. A high-affinity iron permease essential for Candida albicans virulence. Science 288: 1062-1064. 

No further comments.

Author Response

The authors have addressed the issues I raised satisfactorily.

I suggest citing the Science paper published by R Narendrakumar et al. that first reported the discovery of FTR1 to replace Ref. 36. A high-affinity iron permease essential for Candida albicans virulence. Science 288: 1062-1064.

Response: We are deeply grateful for the recognition of reviewer. We agree with the suggestion of reviewer, and have cited the FTR1 paper recommended by reviewer In the revised manuscript as follows:

“36. Ramanan, N.; Wang, Y., A high-affinity iron permease essential for Candida albicans virulence. Science 2000, 288, (5468), 1062-1064.”